# A Knowledge Graph-based Approach for Situation Comprehension in Driving Scenarios

Lavdim Halilaj[1]✉, Ishan Dindorkar[2], Jürgen Lüttin[1], and Susanne Rothermel[1]

[1] Bosch Corporate Research, Renningen, Germany
{lavdim.halilaj,juergen.luettin,susanne.rothermel}@de.bosch.com
[2] Robert Bosch Engineering and Business SPL, Bengaluru, India
{ishan.dindorkar}@in.bosch.com

**Abstract.** Making an informed and right decision poses huge challenges for drivers in day-to-day traffic situations. This task vastly depends on many subjective and objective factors, including the current driver state, her destination, personal preferences and abilities as well as surrounding environment. In this paper, we present *CoSI* (Context and Situation Intelligence), a Knowledge Graph (KG)-based approach for fusing and organizing heterogeneous types and sources of information. The KG serves as a coherence layer representing information in the form of entities and their inter-relationships augmented with additional semantic axioms. Harnessing the power of axiomatic rules and reasoning capabilities enables inferring additional knowledge from what is already encoded. Thus, dedicated components exploit and consume the semantically enriched information to perform tasks such as *situation classification*, *difficulty assessment*, and *trajectory prediction*. Further, we generated a synthetic dataset to simulate real driving scenarios with a large range of driving styles and vehicle configurations. We use KG embedding techniques based on a Graph Neural Network (GNN) architecture for a classification task of driving situations and achieve over 95% accuracy whereas vector-based approaches achieve only 75% accuracy for the same task. The results suggest that the KG-based information representation combined with GNN are well suited for situation understanding tasks as required in driver assistance and automated driving systems.

**Keywords:** Situation Comprehension · Knowledge Graph · Knowledge Graph Embedding · Graph Neural Network.

## 1 Introduction

Safe driving requires an understanding of the current driving situation which includes perceiving the current traffic situation, comprehending their meaning and predicting what could happen in the near future. Situation Awareness (SA) is a concept that attempts to describe and integrate these cognitive processes [12]. SA is the driver's useful moment-to-moment knowledge and understanding of the driving environment but does not include the decision making. SA by machine could assist the driver and reduce accidents by warning about difficult driving

situations. However, the behaviour of the driver in the vast range of situations is not well understood [7].

Vehicles with driver assistance systems (DAS) [4] aim to take some work-load off the driver to improve comfort and efficiency and to enhance driving safety. These systems are aware of the driving situation and benefit from the concept of SA at different task levels of perception, decision making and action [33]. A shared control driver assistance system based on driver intention identification and situation assessment has been proposed in [28]. The application of driver safety warning, particularly collision warning, has been described in [25].

Automated Driving Systems (ADS) require the system rather than the driver to maintain high safety performance. A number of metrics to define the driving safety performance of ADS and compare it to that of human driven vehicles have been proposed in [42]. Currently, in many highly automated driving scenarios, the driver is required to take-over the driving task in cases where the system is not capable to handle the situation safely [1]. This requires machine perception to assess the current driving situation, followed by scene understanding and decision making. Whereas much progress has been made in machine perception, scene understanding and prediction of the next actions of the traffic participants is still subject to extensive research [10].

In this paper, we propose CoSI, a Knowledge Graph (KG)-based approach for representing numerous information sources relevant for traffic situations. It includes information about driver, vehicle, road infrastructure, driving situation, and interacting traffic participants. We built an ontology to encapsulate the core concepts crucial for the driving context. Concepts from external ontologies are reused, enabling an easy extension and interlinking with different data sources, as well as facilitating data extraction. We describe how the knowledge in the KG is utilized via an embedding method such as Graph Neural Networks (GNN) to implement typical classification and prediction tasks used in DAS and ADS. Our approach is evaluated on a synthetically generated dataset comprising a large number of traffic situations and driving styles. We also compare the performance of our proposed approach with classic vector-based feature representations.

## 2 Related Work

Some key tasks in DAS and ADS are the detection and tracking of the relevant traffic participants, prediction of their possible actions, understanding of the traffic situation and planning of the next movement based on the current context. According to [26], approaches for vehicle motion prediction can be grouped into *Physics-based*, *Maneuver-based* and *Interaction-based*.

Classical *Rule-based decision-making* systems in automated driving are limited in terms of generalization to unseen situations. Deep reinforcement learning (DRL) is therefore used to learn decision policies from data and has shown to improve rule-based systems [29]. Different deep neural approaches and feature combinations for trajectory prediction are described in [27], in which surrounding vehicles and their features are extracted from fixed grid cells. Convolutional neu-

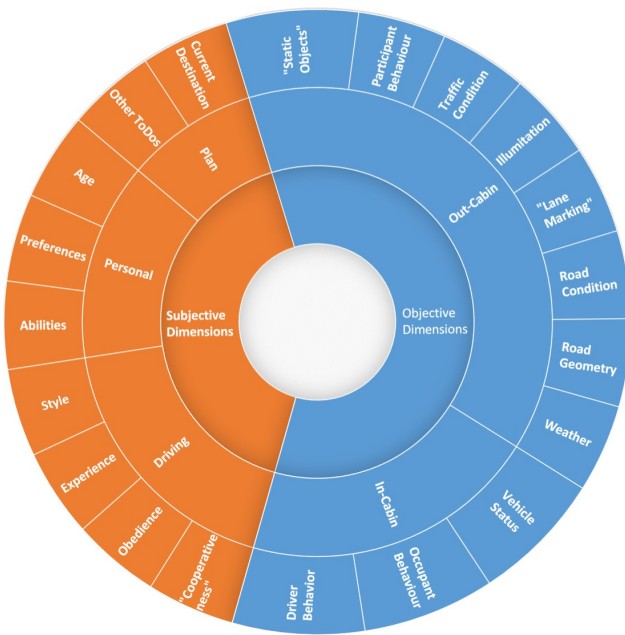

Fig. 1: **Driving Dimensions**. Categories of driving dimensions: 1) *Objective Dimensions*, further divided in Out-Cabin and In-Cabin; and 2) *Subjective Dimensions*, grouped in Personal, Driving, and Plans.

ral networks (CNN) that use fixed occupancy grids are limited by the grid size and the number of vehicles considered [30]. Recurrent neural networks (RNN) are able to model temporal information and variable input size, but are not well suited to handle a variable number of objects permutation-invariant w.r.t. the input elements [20]. When RNNs are combined with an *Attention* mechanism, they can be used to create a representation which is permutation-invariant w.r.t. the input objects [38]. *Deep Sets* provide a more flexible architecture that can process inputs of varying size [43] and are used for DRL in Automated Driving [20]. One limiting factor of these approaches is the implicit and informal representation of entities and relational information between entities.

*Ontologies* on the other hand encompass the formal definition of entities and their relations. Authors in [6] present an approach that uses ontologies to represent knowledge for driver assistance systems. A traffic intersection description ontology for DAS is described in [21]. It also uses logic inference to check and extend the situation description, and to interpret the situation, e.g. by reasoning about traffic rules. An ontology-based driving scene modeling, situation assessment, and decision making for ADS is proposed in [19].

*Graph Neural Networks (GNN)* have been applied to model traffic participant interaction[11, 8]. VectorNet, a hierarchical graph neural network is used for behaviour prediction in traffic situations [15]. A behaviour interaction network that captures vehicle interactions has been described in [13].

Our approach in comparison is overarching, consisting of phases for perception, knowledge ingestion, and situation comprehension. It exploits relationships between the ego and foe vehicles from a graph-based representation. Personal and subjective aspects of the driver and other involved participants are covered as well. Our motivation is that explicit encoding of information might lead to improved modelling of interactions between vehicles. We use multi-relational graph convolution networks [41] that are able to encode multi-modal node features.

## 3    Driving Dimensions

Assisting the driver with fully-informed decisions about steering, acceleration, braking, or more complex tasks like lane changing requires a high level of situation understanding. Figure 1 illustrates an overview of dimensions divided in two main groups, namely *Objective Dimensions* and *Subjective Dimensions*.

**Objective Dimensions**  There exist a number of dimensions that can be explicitly perceived from sensors or derived via specific methods. This group comprises dimensions presented in [16], which are related to dynamics, complexity, and uncertainty impacting the drivability of a driving scene irrelevant of the driver's personality. The ability to scrutinize them in a right manner directly influences the comprehension of the occurring events. Potential hazardous situations can only be identified by further investigating the interactions and the intent of road participants [36]. This information is rather implicit and has to be inferred by combining observations with additional algorithmic procedures.

Considering the origin of these dimensions, they are divided into two categories: 1) *Out-Cabin*; and 2) *In-Cabin*. The *Out-Cabin* category comprises information happening outside the vehicle boundaries. Such information include the actual traffic, road and weather conditions, static objects, illumination, and others. All kind of information pertaining to what is happening inside the vehicle belongs to the *In-Cabin* category. It consists of information related to driver status, occupant behavior, vehicle status, and others.

An exhaustive list of the objective dimensions including their categorization and further details which impact the drivability of a scene are presented in [16].

**Subjective Dimensions**  Objective dimensions alone are not sufficient for a personalized situation assessment. It heavily depends on the individuality of the driver itself. Not each explicitly perceived situation occurring in a given driving scenario is considered the same or has the equal level of difficulty. Therefore, a number of subjective dimensions are crucial in determining correct situation classification or difficulty level. The subjective subcategories cover information pertaining to *personal*, *driving*, and *current plans*.

In general, each subcategory belonging to the outer circle can have tens of signals coming from sensors through different transmission channels like Controller Area Network (CAN) bus. These signals include information such as gaze direction, drowsiness, inside- and outside-temperature, fuel consumption, number of occupants. Next, special processing units consume and manage retrieved signals to generate appropriate notifications to the driver or actions to the vehicle.

# 4 Approach

Achieving a *high level of intelligence* is possible by fusing and enriching collected data from different sources such as connected sensors. This enables autonomous vehicles to react according to the situation within a driving environment [9]. With the aim of covering the entire process, we designed a flow-oriented architecture composed of three main phases, namely: 1) *Contextual Observation*; 2) *Knowledge Ingestion*; and 3) *Situation Comprehension*. Each phase comprises a number of components dedicated to perform specific tasks as shown in Figure 2.

## 4.1 Contextual Observation

Intelligent vehicles are equipped with a number of sophisticated sensors that sense the surrounding of the vehicle. Typically, the average number of sensors in a smart vehicle is ranging from 70 up to 100 [18], monitoring various types of events as well as stationary and mobile objects. This includes observing the driver and occupant(s), engine status, and the road network. Spatial and temporal information is obtained for each observed object. A wide range of sensors, e.g. light and rain sensors, internal and external cameras, Radar and Lidar, are used for perception tasks. Each sensor may be built via a specific technology and standard, thus generating data in various formats with a different granularity.

## 4.2 Knowledge Ingestion

Situation comprehension requires integrating and structuring the abundance of information from various sources. Raw signals from sensors are transformed and enriched with additional semantics. Further, contextual information and user characteristics are injected to support a personalized situation assessment.

**Knowledge Graph** The KG serves as a coherence component comprising fundamental ontologies to capture information about entities and their relationships. We see a KG as a set of triples $G = H, R, T$, where $H$ is a set of entities, $T \subseteq E \times L$, a set of entities E or literal L values and $R$, a set of relationships connecting $H$ and $T$. These triples are represented using Resource Definition Framework (RDF) as a modeling language. Encoding additional formal axioms enables inferring new facts out of given ones via automated reasoning techniques.

Once the transformation process is realized, i.e. converting input data of any format to triples, the output is stored in a knowledge graph. Information in the KG is aggregated and organized in an intuitive and hierarchical way, making it easy to exploit and understand by humans. An excerpt of the CoSI KG (CKG) is given in Figure 3, showing how *scenery information* is represented via instances (*assertional box*) of ontological concepts (*terminological box*). Apart from sensor data, it captures the information related to the driver, such as preferences and abilities modeled according to the CoSI ontology, as described in the following.

*Ontology* We developed the CoSI ontology based on the dimensions and their respective categories described in Section 3. It captures relevant information coming from sensors mounted in a given vehicle. The human description of crucial concepts: Scene, Situation and Scenario given in [37] are used as a basis to create

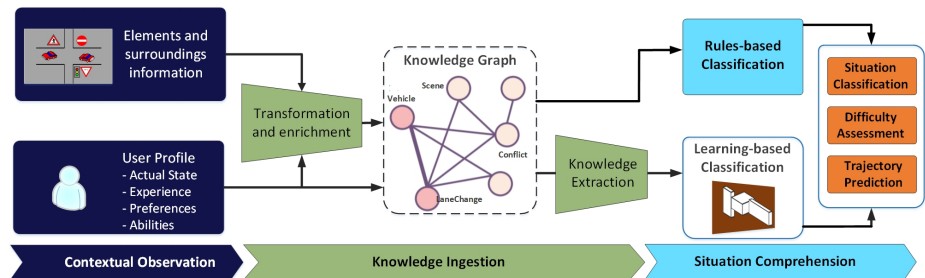

Fig. 2: **CoSI Pipeline**. The pipeline is comprised of three consecutive phases: 1) *Contextual Observation* - capture information about the surroundings and the user; 2) *Knowledge Ingestion* - transform, enrich and ingest information on the KG; and 3) *Situation Classification* - assess the situation type considering the contextual knowledge.

formal definitions using ontological axioms. Additionally, the ontology models different user characteristics, such as preferences (e.g., preferred driving style or safety measures), experience, and (dis)abilities. In the following, the respective definitions of core concepts of the CoSI ontology are given:

– *Scene*: A scene describes a snapshot of the environment including the scenery and dynamic elements, as well as all actors' and observers' self-representations, and the relationships among those entities [37].
– *Situation*: A situation is the entirety of circumstances considered for the selection of an appropriate behavior pattern at a particular point of time. It entails all relevant conditions, options, and determinants for behavior [37].
– *Scenario*: A scenario describes the temporal development between several scenes in a sequence of scenes. Actions, events, and goals may be specified to characterize this temporal development in a scenario [37].
– *Observation*: Act of carrying out an (Observation) Procedure to estimate or calculate a value of a property of a FeatureOfInterest [17].
– *Driver*: A driver is a specific type of user. It encapsulates all relevant attributes associated to a driving context where driver is the main subject.
– *Profile*: A user profile is a structured data representation that is used to capture certain characteristics about an individual user[3].
– *Preference*: A preference is a technical term in psychology, economics and philosophy usually used in relation to choosing between alternatives. For example, someone prefers A over B if they would rather choose A than B[4].

The CoSI ontology is built on principles for an easy extension and exploration. Concepts from external ontologies such as Schema.org and SOSA ontology[5], are reused to enable interlinking with different data sources. Currently, it contains 51 classes, 57 object datatype properties, and 3 annotation properties.

**Transformation and Enrichment** This component performs a semi-automatic conversion of sensor data to the RDF representation via both declarative and

---

[3] https://en.wikipedia.org/wiki/User_profile
[4] https://en.wikipedia.org/wiki/Preference
[5] https://schema.org/, https://www.w3.org/TR/vocab-ssn/

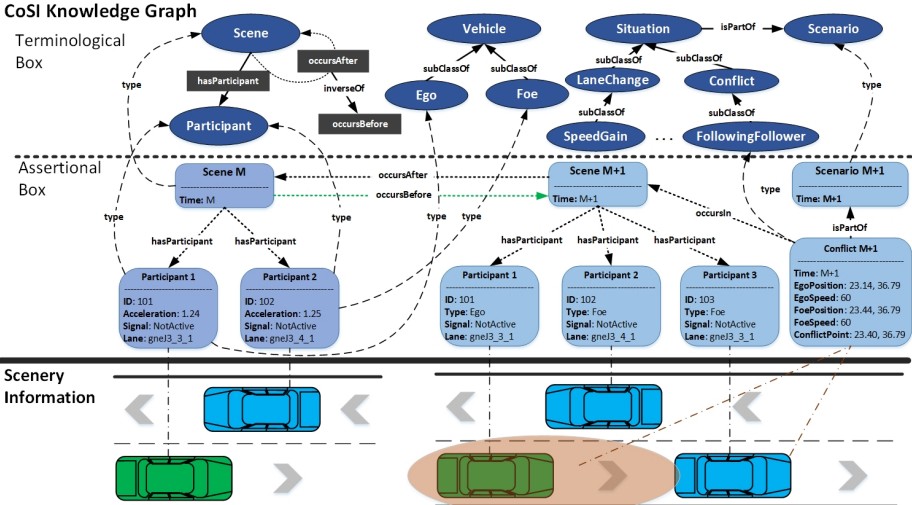

Fig. 3: **CoSI Knowledge Graph**. An excerpt of the CoSI KG representing respective situations occurring in two consecutive scenes: 1) the bottom layer depicts scenery information among participants; 2) the top layer includes concepts such as classes and relationships representing the domain knowledge; and 3) the middle layer contains concrete instances capturing the scenery information based on the ontological concepts.

imperative approaches. Using the declarative approach, a number of mappings of sensor data to the ontological concepts are defined. The imperative approach is realized in cases when it is necessary to perform complex transformations. In these cases, additional queries are executed on-the-fly to enrich sensor data with new relationships. For instance, raw data generated from sensors are augmented with additional semantic information in order to establish new type definitions or missing relationships between scenes, e.g. *occursAfter*.

**Knowledge Extraction** Performing tasks such as knowledge graph completion, link prediction, classification, or other types of downstream tasks requires knowledge to be consumed based on various perspectives. This component allows for execution of complex queries and traversing the graph to retrieve relevant information. Various views of the information can be created on the fly while the underlying knowledge structure remains unchanged. For instance, while originally the information is organized from the perspective of a Scene, i.e. participants, their position, speed, as well as the type of situation happening in it; by traversing the graph through specific queries, another view from the ego vehicle perspective can easily be generated as illustrated in Figure 4. As a result, embedding techniques that operate on graph level, can efficiently learn the vector representation of symbolic knowledge from the "new perspective".

### 4.3 Situation Comprehension

A number of specialized components perform dedicated tasks related to situation comprehension, such as *Situation Classification*, *Difficulty Assessment*, and

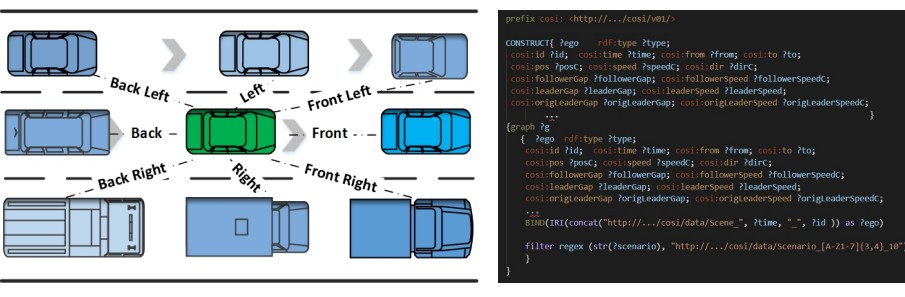

(a) Ego View  (b) Construct Query

Fig. 4: **Ego Vehicle Perspective**. a) A traffic situation from the ego vehicle (green) perspective in the center and eight foe vehicles in its vicinity. b) A SPARQL query constructing a graph-based view on the fly from ego perspective.

Table 1: **Simulation Parameters**. Simulation parameters of the driver and vehicle that were varied to generate a large range of driving situations.

| Parameter | Description |
|---|---|
| lcSpeedGain | Driver's eagerness for speed gain by overtaking |
| lcCooperative | Driver's cooperativeness in reducing speed for other vehicles |
| $\sigma$ (sigma) | Driver's imperfection in realizing desired speed |
| $\tau$ (tau) | Driver's reaction time in seconds |
| minGap | Driver's target gap between ego and foe vehicle |
| maxSpeed | Vehicle's maximum speed |
| maxAccel | Vehicle's maximum acceleration |

*Trajectory Prediction.* These components can be built on different paradigms, namely 1) Rule-based; and 2) Learning-based classification.

- *Rule-based Classification* Components following this paradigm rely on a number of declarative rules to perform logic-based classification tasks. They harness the expressive power of the knowledge graph structure which in combination with reasoning techniques provide interpretable results.
- *Learning-based Classification* Components employ a number of techniques that learn common patterns from a large number of observations. Therefore, it is possible to make predictions or classifications based on a given sample data used for training without pre-defining explicit rules.

## 5 Implementation

Our objective is to evaluate how well our KG-model can represent traffic situations. Particularly, we focus on situations with interacting vehicles, where vehicles base their behaviour on current and predicted behaviour of other nearby vehicles, considered a notoriously challenging task. We thus define driving situation classification as our evaluation task. To validate the benefit of our model with as little influence from other factors, we deliberately define the task on single *scenes*. Models that exploit temporal information are very likely to lead to better results, since some situations such as merging or overtaking are manifested via a gradual change of the lane ID over time. However, here we only validate the underlying static knowledge and exclude effects of temporal information.

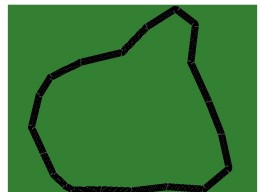 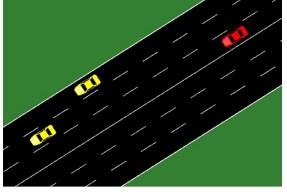 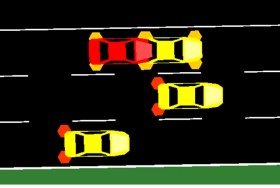

(a) Gothenburg's Highway    (b) Lane change situation    (c) Collision situation

Fig. 5: **Sumo traffic simulations**. Examples of traffic simulations using SUMO: a) A snapshot of a highway around the city of Gothenburg; b) A situation where ego vehicle is switching the most left lane; and c) An example with a collision situation happening.

### 5.1 Dataset Generation

**Experimental Setup** Experiments on recorded test drives is prohibitive [22] as it would require huge amounts of test data without assurance that enough examples of critical driving situations are included. We therefore use simulated data for our experiments. An advantage of this approach is that it allows to specifically generate critical driving scenes which we are interested in, both in terms of driving behaviour and situation criticality.

We use *Simulations of Urban Mobility* (SUMO)[6], an open source, highly portable, microscopic and continuous multi-modal traffic simulation package to generate driving data. We generated a dataset[7] comprising more than $50'000$ driving examples on a 40 km long highway section around the Gothenburg city as shown in Figure 5a. It is circular in shape with three lanes in each direction.

*Scenarios* We varied simulation parameters such as time-to-collision (TTC) (i.e. the time until a collision between two entities would occur if both continue with the present velocities) of drivers and vehicles to simulate different driving styles and vehicle types as listed in Table 1. This leads to the generation of various driving scenarios as described in Table 2.

*Car Following Model* The standard car following model in SUMO is described in [24]. It is based on the assumption that the speed of the ego vehicle is adapted according to the speed of the preceding vehicle. Further, a desired gap between the ego and the leading vehicle as well as reaction and braking time are taken into account to ensure that no collision will occur. Figure 5c depicts a situation where a collision between the ego and the foe is happening.

*Lane Change Model* The lane change model considers different motivations of the driver (route, speed gain, rule-following, cooperativeness and other factors) for lane changes on multi-lane roads and related speed adjustments [14]. Figure 5b shows an example where the ego vehicle is performing a lane change.

### 5.2 CoSI Knowledge Graph

Data generated from SUMO are provided in XML format which are then transformed and enriched on-the-fly to RDF representation. As a result, the CoSI

---

[6] https://www.eclipse.org/sumo/

[7] https://github.com/siwer/Retra

Table 2: **Driving Scenarios**. Different driving scenarios are simulated by varying simulation parameters of vehicle and the driving style.

| Driving scenario | Description |
| --- | --- |
| 1 | Successful lane change |
| 2 | Abandoned lane change |
| 3 | Dangerous lane change (small gap, small TTC values) |
| 4 | Dangerous close car following (small gap, small TTC values) |
| 5 | Unexpected stopping of leading vehicle |
| 6 | Unexpected pedestrians on road |
| 7 | Collision between ego and foe vehicle |

Knowledge Graph (CKG) is created with over 915 million triples. There are more than 84K *Conflict Scenarios* containing millions of instances of different types of situations, such as *Following Leader*, *Following Follower*, *Crossing*, *Merge* etc. These instances consist of information about the conflict points, speed of ego and foe vehicle, and the direction of movement, respectively. Further, CKG has more than 10K *Lane Change Scenarios*, which also contains millions of instances of different lane change situations, categorized in respective ontological classes, such as *Speed Gain*, *Keep Right*, or *Sublane*.

### 5.3 KG-based Situation Classification

This component is implemented according to the principles of the learning-based paradigm. It uses a *Relational Graph Convolutional Network* [34] (R-GCN), an extension of *Graph Convolutional Network* (GCN) [23], to directly operate on a graph and learn its embeddings. Each layer $l$ of the *R-GCN* calculates:

$$H^{(l+1)} = \sigma(\sum_{r\in\mathbb{R}} \hat{A}^r H^{(l)} W^{(l)})$$ (1)

where $\hat{A}^r$ is the normalized adjacency matrix of the graph $G$, $W^{(l)}$ is the weight matrix for layer $(l)$ and $\sigma$ the activation function, such as *ReLU*. $H^{(l)} \in \mathbb{R}^{N\times D}$ is the matrix of activation's in the $l^{th}$ layer and $H^0$ the $N \times D$ matrix of $D$-dimensional node embeddings for nodes $N$ in the graph. Similar as in [41], we extend the matrix of node embeddings $H$ with feature embeddings for each node, forming a *Multimodal Relational Graph Convolutional Network* (MRGCN):

$$H^{(l+1)} = \sigma(\sum_{r\in\mathbb{R}} \hat{A}^r H_I^{(l)} W_I^{(l)} + \hat{A}^r H_F^{(l)} W_F^{(l)})$$ (2)

where $W_I^{(l)}$ and $W_F^{(l)}$ are the learnable weights for the structural and feature components, respectively.

### 5.4 Vector-based Situation Classification

To compare the performance of the KG-based classification with traditional feature based classifiers we implemented three methods described below. The features are extracted from our KG to represent vector-based features. To deal with

a varying number of foe vehicles, we formed samples with tuples of the ego vehicle, a foe vehicle and their relational information such as TTC and distance.

- *Support Vector Machine (SVM)* SVMs are a set of supervised learning methods that use a subset of the training samples, the so-called support vectors, in the decision function.
- *Decision-Tree Classifier (DTC)* The fact that we have both, numerical and categorical features representing driving situations, the application of decision tree fits well to our classification task.
- *Multi-Layer Perceptron (MLP)* Finally, we use a multi-layer perceptron neural network and train in with back-propagation for the classification task.

## 6 Evaluation

We empirically study the accuracy of our knowledge graph-based approach in a situation classification task. In this section, we describe in detail the experiment configuration for both, the KG-based as well as for the vector-based approaches, as well as the achieved results. To evaluate the performance of our KG-model, we compare classification results with vector-based feature representation in combination with different classifiers.

### 6.1 Experiment Configuration

For our experiments, we used a sub-set of the CKG with 6.4 million triples, representing around 226K Conflict Situations, each described with 28 triples on average. This is further divided in 134K for training, 46K for validation and 46K for testing. To investigate the relative importance of different features, we performed experiments with single features and combined features as shown in Table 3. The 5 most important features were selected based on heuristics. For each vehicle we have about 10 features on average (e.g., position, speed, acceleration, steering angle). The number of other features (e.g., minTTC, velocity difference, etc.) depends on the number of vehicles in the vicinity (50m radius) around ego. The numeric features are normalized before further processing by the classifiers. MRGCN is implemented using one hidden layer with 40 nodes[8], trained in full batch mode with an ADAM optimizer. MLP uses 2 hidden layers with 40 hidden neurons per layer trained in full batch mode with an ADAM optimizer. SVM is implemented with radial basis function kernels and DTC uses the *Classification And Regression Trees* algorithm.

### 6.2 Results and Discussion

Results for the different methods and different number of features are shown in Table 3. The overall best results are obtained by the MRGCN method using the 5 most important features. For MRGCN, the performance using all features is slightly lower than for the 5 most important ones. For the vector-based methods SVM and DTC, best results are obtained using all features. When only one feature is used, DTC and MLP achieve the best performance.

---

[8] https://gitlab.com/wxwilcke/mrgcn

Table 3: **Classification accuracy**. Accuracy for different features and different algorithms for the task of situation classification.

|  | MRGCN | SVM | DTC | MLP |
|---|---|---|---|---|
| Single Features |  |  |  |  |
| Vehicle signal | 0.468 | 0.510 | **0.511** | **0.511** |
| Longitudinal lane position | 0.501 | 0.566 | 0.507 | **0.569** |
| Steering angle | 0.607 | 0.524 | **0.641** | 0.510 |
| Distance between vehicles | **0.673** | 0.529 | 0.491 | 0.542 |
| Lane ID | 0.883 | 0.520 | **0.890** | 0.514 |
| Combined features |  |  |  |  |
| ALL 36 features | **0.938** | 0.708 | 0.750 | 0.733 |
| 5 most important features | **0.953** | 0.648 | 0.746 | 0.742 |

The experiments show that our KG-based classifier achieves considerably better results than all vector-based classifiers, in case when 5 most important or all features are used, respectively. This suggests that the graph-based representation provides more discriminating information compared to the classic feature vector-based representation.

Classification experiments with single features show that vector-based methods are superior to KG-based classifier for all features except *Distance between vehicles*. This suggests that KG-based methods have no clear advantage in simple situations but outperform vector-based methods in complex situations where multiple relationships and interactions exist between participants. It indicates that KG-based methods could also perform well in more complex tasks that consider much rich context as well as domain knowledge about the driver.

We did not consider temporal information for our classification experiment. We believe that a superior performance of the KG-based method in learning relational information suggests that the method will also be able to learn temporal relations between nodes. The conducted experiments were the first attempts to prove the advantage of our CoSI approach and further experiments for tasks in automated driving will be the subject of future research.

## 7 Usage and Lessons Learned

### 7.1 Usage

Bosch has been pushing research in automated driving for many years and as a result, many technologies are ready for highly automated driving today [5]. For example, Bosch has developed automated valet parking, the first fully automated system (SAE level 4). Therefore, the approach described here is one of the many ongoing activities to address the challenges in automated driving [3]. It represents one component of the overall architecture for an autonomous driving system [2].

The behavior of highly automated driving (HAD) systems especially in critical driving situations is crucial for their validation [39]. However, the validation based on recorded test drives would require millions or billions of test kilometers which makes it unfeasible [22]. Thus, alternate methods of validation including

simulation are a common practice [44]. We therefore used simulated data to validate our approach. On the other hand, HAD vehicles will contain sensors and perception units providing information about the driving scene surrounding the ego-vehicle, similar to the information provided by the simulation tool (e.g. position, speed, and driving direction of nearby vehicles). We therefore expect that our approach is generic enough and will also perform well on real-world data.

### 7.2 Lessons Learned

**Maturity of semantic technologies** For many years now, semantic technologies are widely used in specific domains e.g. education, life sciences and cultural heritage [31]. Recent advances of vendors such as Stardog, OntoText, and Cambridge Semantics[9] on their respective solutions offer support for many use cases with different requirements and scenarios. Their primary function as triple stores is improved considerably, being now able to manage and query knowledge graphs with trillions of triples without experiencing significant degradation in performance. Other features such as knowledge exploration, visualization, or validation techniques are now inseparable and fully integrated in many triple stores. A wide range of industry related solutions can be implemented and fully operate in production, instead of a prototypical level. Therefore, we also consider their application in the automotive domain to be promising.

**Integration with existing data sources** Typically, the sophisticated triple stores provide support for the *Ontology-Based Data Access* (OBDA) principles [32]. OBDA enables accessing heterogeneous data sources via ontologies and a set of mappings, that interlink this data with the ontological concepts.

There are two main forms for realizing a data integration scenario: 1) *virtual data access* - data are kept in the original format but are transformed on-the-fly and accessed as RDF triples; 2) *materialization of triples* - data from relational tables are materialized in RDF triples to special named graphs. We followed a hybrid approach using both forms depending on the requirements and constraints:

- *Virtual data access* is seen as a preferred solution in cases when it is crucial to avoid: 1) synchronization issues with data frequently changing; 2) replicating resources; and 3) issues with migrating legacy systems.
- *Materialization of triples* is applied to prevent from: 1) performance degradation of running systems while executing complex queries; 2) safety and security issues with read/write permissions; and 3) issues with heavyweight reasoning for non-RDF data.

**Applicability of knowledge graph embeddings** Knowledge graphs are powerful in encapsulating and representing prior knowledge, leveraging rich semantics and ontological structures. Additional rules and axioms encoded manually support the reasoning process where new facts are inferred from the existing ones. On the other hand, a number of knowledge graph embedding (KGE) methods

---

[9] https://www.stardog.com, https://www.ontotext.com, https://www.cambridgesemantics.com

are presented for learning latent information in a KG using low dimensional vectors [40]. They perform tasks such as knowledge graph completion, entity recognition and classification, as well as *downstream tasks* like recommendation systems, question answering, and natural language processing.

Despite the fact that from their intrinsic nature, knowledge graphs are very flexible, we faced a number of challenges while preparing the data to be optimally processed by KGE methods. In particular, we had to define special queries each time when the information is scattered in n-ary relationships more than one hop away from the main node. A number of slightly different views are created on the fly comprising additional information encapsulated in main nodes. As a result, the KGE methods performed better in terms of achieving a higher accuracy.

## 8 Conclusions

This article presents CoSI, an approach for enabling situation comprehension using knowledge graphs. The CoSI ontology as the skeleton of CKG provides a semantic representation of core concepts in a driving context. Thus, our approach is able to effectively integrate data from heterogeneous sources and structures it into a common knowledge graph. To demonstrate the applicability of our approach, we performed a number of empirical evaluations with different machine learning methods. Results show that our approach achieves higher accuracy in the task of situation classification compared to traditional methods. This indicates that CKG can well represent complex information of the driving domain and when combined with graph-based neural networks leads to superior performance, achieving over 95% accuracy.

As future work, we plan to further expand the CoSI ontology with more fine-grained entities covering additional objective and subjective dimensions. Next, we will complement our approach via implementing the classification based on axiomatic rules described in [35]. Therefore, tasks such as *difficulty assessment* and *trajectory prediction* can be performed following *rule- and learning-based* paradigms, respectively. In order to improve tasks related to situation comprehension, we will further exploit objective dimensions, i.e. related to the context and time. Another goal is to include subjective dimensions in KGE methods for achieving a more personalized situation comprehension.

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
