# OpenReview forum: "A Knowledge Graph-based Approach for Situation Comprehension in Driving Scenarios"
_eswc-conferences.org/ESWC/2021/Conference/In-Use_Track — ESWC 2021 In-Use_

### Official Review · AnonReviewer2 · 2021-01-14
**Comprehensible approach with the promising results**

**Rating:** 1
**Confidence:** 3

**Review:**

The paper presents the knowledge-based approach for understanding the driving scenario. The approach consists of 3 steps: contextual observation, knowledge ingestion, and situation classification. The CoSI ontology is developed to use for expressing the situation as a knowledge graph. The empirical experiment is conducted to show the effectiveness of the approach. Overall, the paper is very interesting and the results are promising.

Detail comments
- In Fig 1, Why were these dimensions chosen? Do other dimensions exist? The text lists most of what was included but does not argue why something was included/excluded. Is this set of dimensions meant as an example or is it proposed as a comprehensive list of what is useful for ADS/DAS? For example, we have [Subjective->Personal->Age] as well as [Subjective->Personal->Abilities, Subjective->Driving->Style, Subjective->Driving->Experience]? Does Age provide some unique information that is not already included in the other three?
- The division between the original work and citing [14] in creating the list of dimensions is not clear in either the text (Sections 3 and 3.1) or Figure 1.
- (Page. 9 5.1) "We use a dataset of simulated driving situations as this allows to generate a large range of driving behavior and traffic situations compared to the use of recorded driving scenes." This seems to be true at least from the point-of-view of practicality, but is this true in general? A huge amount of recorded driving data is available, although it is difficult and very time-consuming to process into uniform and usable format. The scenarios in Table 2 list what seems to be one common case (#1 successful lane change) while the rest are more uncommon scenarios. Simulation is a reasonable approach to include a good number of varying uncommon situations while avoiding ending up with a huge amount of data that is sparse in terms of situations of interest. The problem of this sentence boils down to it being a very general statement without argumentation.

Small comments
Grammar should be comprehensively checked as many problems exist. Some, but not all, examples:
- Abstract: "...Harnessing the power of axiomatic rules and reasoning capabilities, enables inferring additional knowledge..."-> Remove comma
- Pg. 2 Introduction: "...take some work-load of the driver..." -> "of" should be "off" and "Shared control driver assistance system based on driver intention identification and situation assessment have been proposed in [25]." -> "have" should be "has"
- Pg. 4 Section 3.2: "The following subjective subcategories cover information pertaining to personal, driving, and actual plans." -> remove the word "following"
- Pg. 5 Section 4.1: "Comprehending situation requires integrating and structuring the abundance of information from different sources before being ready to be used."  -> "Comprehending a situation requires integrating and structuring the abundant information from different sources before it is ready to be used"
- Fig 1. "Illumitation" -> "Illumination" Empty slot on outer ring?
- Fig 2.: Transformation and enrichment is included in ingestion, however, the caption states the following: "2) Knowledge Ingestion - transforming and enriching information before ingestion on the Knowledge Graph;". Either this is contradicting itself or "ingestion" is used to describe two different things in one sentence.

**Anonymity:**

Yes, I would like my review to remain anonymous.

**Subreviewer:**

I submitted this review.

---

> ### Author Rebuttal · Authors · 2021-01-28
>
> Thank you for your constructive review. We are pleased to provide our answers to open points:
>
> 1.	In Fig 1, Why were these dimensions chosen? Do other dimensions exist? The text lists most of what was included but does not argue why something was included/excluded. Is this set of dimensions meant as an example or is it proposed as a comprehensive list of what is useful for ADS/DAS? For example, we have [Subjective->Personal->Age] as well as [Subjective->Personal->Abilities, Subjective->Driving->Style, Subjective->Driving->Experience]? Does Age provide some unique information that is not already included in the other three?
>
> We agree that the list of dimensions is not exhaustive and further dimensions exist. A major objective for us was to categorize these dimensions considering their origin and perspective. In the combination with findings from the state of the art, in this work we focused on some more representative ones, and exploring the possibility of future extension. In particular, that is going to be more relevant when we will be able to access and work with real sensor data coming from various sources like Controller Area Network (CAN bus).
>
> Regarding the age and other similar dimensions, here we considered various requirements applied in different countries w.r.t. driving policies. For instance, countries may (now or even in the future) apply restrictions based on age, like driving only on specific roads, up to some speed limit, or using only automatic gear transmission, etc.
>
> 2.	The division between the original work and citing [14] in creating the list of dimensions is not clear in either the text (Sections 3 and 3.1) or Figure 1.
>
> We agree that the distinction between our work and the [14] should be more explicitly stated. The dimensions listed in [14] belong to the category that we classified as “Objective Dimensions”, i.e. dimensions related to what is happening inside and outside cabin. In addition, we extended the list with the category of “Subjective Dimensions” mainly related to the driver personality and characteristics.
>
> 3.	(Page. 9 5.1) "We use a dataset of simulated driving situations as this allows to generate a large range of driving behavior and traffic situations compared to the use of recorded driving scenes." This seems to be true at least from the point-of-view of practicality, but is this true in general? A huge amount of recorded driving data is available, although it is difficult and very time-consuming to process into uniform and usable format.
>
> Highly automated driving of at least ASIL Level 3 is of  the research focuses of future cars (Koopman 2016). The behavior of HAD especially in critical driving situations is of particular importance in order to validate HAD systems and compare their performance with human drivers (Wang 2019). Validation based on recorded test drives (Waymo 2017) would require hundreds of millions or hundreds of billions of test kilometers to prove safety of HAD systems which makes them unfeasible for validation. Virtual testing is one main method to validate HAD systems (Wang 2019). Sumo is one of the main simulation tools for traffic simulation (Lopez 2018).
>
> (Kalra 2016) N. Kalra and S. M. Paddock, “Driving to safety: How many miles of driving would it take to demonstrate autonomous vehicle reliability?,” Transportation Research Part A: Policy and Practice, vol. 94, pp. 182–193, 2016.
>
> (Koopman 2016) P. Koopman and M. Wagner, “Challenges in Autonomous Vehicle Testing and Validation,” SAE International Journal of Transportation Safety, vol. 4, no. 1, pp. 15–24, 2016.
>
> (Lopez 2018) Lopez, Pablo Alvarez, et al. "Microscopic traffic simulation using sumo." 2018 21st International Conference on Intelligent Transportation Systems (ITSC). IEEE, 2018.
>
> (Waymo 2017)  “On the Road to Fully Self-Driving: Waymo Safety Report,” https://goo.gl/7HUiew, 2017.
>
> (Wang 2019) C. Wang, H. Winner, “Overcoming Challenges of Validation Automated Driving and Identification of Critical Scenarios”, IEEE Intelligent Transportation Systems Converence, 2019.
>
>
> We appreciate the spotted syntactic issues; we will fix them in the last version and will do a thorough check.

---

### Official Review · AnonReviewer4 · 2021-01-14
**An interesting use case applying GNN on top of a knowledge graph**

**Rating:** 1
**Confidence:** 2

**Review:**

The paper presents a knowledge graph-based approach for the situation classification in automated driving scenarios. The authors define an ontology to capture information about both external (observed) situation-related concepts as well as those related to the driver profile and goals. The knowledge graph integrates information coming from sensors with static knowledge. Then, the information is used for situation classification using a combination of rule-based and machine learning techniques. Evaluation experiments compared the classification accuracy with several baselines and showed the improvements in accuracy.

Overall, the topic of using semantic web data for situational awareness is a relevant one and long-studied in the community. The authors’ approach is an interesting one as it adds on top the use of state-of-the-art ML methods, such as graph neural networks.

One aspect that is not clearly discussed in the paper is scalability that is crucial in such real-time scenarios as driving. The “Lessons learned” section briefly mentions the experience with using existing solutions and seems to imply that the triple stores are already scalable enough for this scenario, while OBDA integration might present a bottleneck. Is this the assessment? It would be nice to have a section discussing in more detail the scalability requirements and whether the proposed pipeline can already meet them.

I would like to thank the authors for clarifying these aspects in the rebuttal comments. It seems, though, that data retrieval time as it was measured is not suitable for real-world situations, however, it's understandable that this was not the focus of work.

**Anonymity:**

Yes, I would like my review to remain anonymous.

**Strong Points:**

* Relevant practical use case
* An interesting combination of knowledge graph technologies with novel GNN methods

**Subreviewer:**

I submitted this review.

**Weak Points:**

* Would be nice to have more discussion on scalability and processing time, as the use case is time-critical

---

> ### Author Rebuttal · Authors · 2021-01-28
>
> Thank you for your review and very interesting point! We will detail our view about this very important aspect in the following:
>
> 1.	One aspect that is not clearly discussed in the paper is scalability that is crucial in such real-time scenarios as driving. The “Lessons learned” section briefly mentions the experience with using existing solutions and seems to imply that the triple stores are already scalable enough for this scenario, while OBDA integration might present a bottleneck. Is this the assessment? It would be nice to have a section discussing in more detail the scalability requirements and whether the proposed pipeline can already meet them.
>
> From our perspective OBDA are approaches that 1) use ontologies as basis or 2) map to some ontological concepts data coming for heterogeneous sources. This can be either used for 1) materialization to the RDF triples, or 2) virtual transformation, where data are converted on-the-fly. We see both that approaches which follow set of predefined mappings can be used to best match to the requirements or constraint on place.
> We fully agree with the suggestion about the scalability and the processing time, which are especially important in time-critical systems. However, the subject of this study was more focused on integrating and exploiting the data and performing dedicated tasks but more in the offline mode, where machine learning models are then trained on the extracted data and then used for classification purposes.
> From the technical point of view: we used Stardog (version 7.3.3) as a triple store. Retrieving the scenery information (with two hops and some variable bindings) over a 72 million named graph (we separated the CoSI Knowledge Graph in different named graphs based on different Driving Scenarios generated by SUMO) returned ~163K rows within 29 seconds. This time value (29sec) can be further decreased by doing some optimization of the query, as suggested in Stardog documentation. In addition, we have seen that in the new versions of Stardog, there are additional improvements on query execution techniques, which would increase the speed.
>
> Regarding computational complexity, the current approach using all features takes about 44 minutes for training on 134’000 samples and 11 seconds for testing of 46’000 samples, resulting in 0,239 msec test time per sample.
> The next challenging steps are considering various aspects of the solution including time requirements, privacy requirements, processing capacities, and online learning, which we will investigate deeper in future work.
>
> The next challenging steps are considering various aspects of the solution including time requirements, privacy requirements, processing capacities, and online learning, which we will investigate deeper in future work.

---

### Official Review · AnonReviewer1 · 2021-01-15
**A good paper to show a use case of knowledge graph**

**Rating:** 1
**Confidence:** 3

**Review:**

Pros:

- construct knowledge graph-based framework for driving situation comprehension

- test the framework using simulated data and show the effectiveness of the knowledge graph approach

Cons:

- Ontology and knowledge graph are not in public

- the method itself is not so novel

- some points are not clear

comments:

What is the purpose of the discussion in section3? Is it used for the design of CoSI ontology?

How many situations did you use for this work and how do you create ground truth data? If you use only for classifying collision situation or not by using features, I cannot understand the experimental result. Because it shows that we can predict the situation using Lane ID only with 89% accuracy. Maybe I misunderstand something.

P.5 Sec.4.2
An excerpt of the CoSI KG (CKG) is given in Figure 3,
-> An excerpt of the CoSI KG (CKG) is given in Figure 2,


**Anonymity:**

Yes, I would like my review to remain anonymous.

**Subreviewer:**

I submitted this review.

---

> ### Author Rebuttal · Authors · 2021-01-28
>
> We thank the review for the constructive feedback. Below, we try to answer each of the open points:
>
> 1.	What is the purpose of the discussion in section3? Is it used for the design of CoSI ontology?
>
> Indeed, the discussion in the section 3 serves as a basis for defining the CoSI ontology and to pave the way for further extending it. The categories defined in this section as well as the dimensions provided us a better view on the aspects that we have to consider as well on modelling concepts and their inter-relationships. We will make this clear in the last version of the paper.
>
> 2.	How many situations did you use for this work and how do you create ground truth data?
>
> We used the situation types provided by SUMO. Out of 21 situation types our simulation data consisted of 5 different types (FolllowingFollower, FollowingLeader, OnAdjacentLanes, Collision, NoConflict). The ground truth is provided by the simulation tool.
>
> 3.	If you use only for classifying collision situation or not by using features, I cannot understand the experimental result. Because it shows that we can predict the situation using Lane ID only with 89% accuracy. Maybe I misunderstand something.
>
> For each scene, we classify the driving situation out the classes described above based on the vector based data representation or the CKG based representation.
> Lane ID plays an important role when identifying if a vehicle is following or being followed by another vehicle on the same lane and also if a vehicle is on a neighbouring lane. Using only this feature alone therefore already leads to high performance.
>
>
> We will fix the mistake.

---

### Official Review · AnonReviewer3 · 2021-01-15
**KG and GNN applied to the ITS domain**

**Confidence:** 3

**Review:**

Set in the domain of Intelligent Transport Systems, the paper proposes a KG based approach to semantically annotate and classify different traffic situations.

The paper is well motivated and the use of SW technologies is clear. For the semantic annotation, authors proposed the CoSI ontology. While the ontology reuses parts of well known ontologies, such as Schema.org and SOSA, it seems ontologies on the ITS domain haven't been used. One of the ontologies mentioned in the related work (reference 17) seems to be relevant, but I couldn't access that work. Also, there has been work that suggested an ITS ontology based on the idea of Local Dynamic Map, that might be relevant (see reference below).

The paper describes a pipeline to observe, ingest and comprehend traffic situations. However, parts of the pipeline lack a more detailed description. For instance, the Knowledge Extraction seem to involve a number of different tasks, and it is not clear how they are used in practice. Also, there are many mentions to reasoning approaches, but I could not find a concrete example where reasoning is being used. The evaluation focuses on the performance of the classification task.

Regarding the evaluation, it compares the GNN approach to vector based methods. The dataset was generated used a traffic simulation tool, SUMO. Although data is syntactic, it can nevertheless simulate real traffic situations with high accuracy, and its understandable that such approaches can not be tested in real testbeds.

It would be interesting to provide a description on how SUMO XML data is mapped to the KG. While quickly looking at the SUMO documentation, I couldn't find some of the concepts from the CoSI ontology. One idea is to include an example of how a scene element is created, given a subset of the XML. Also, is the knowledge extraction step being applied here?

Authors have decided not to take temporal aspects of the data into account. As a first test of the approach it is understandable and acceptable to leave some of the complexity out. However, given that the domain is highly dynamic, it would be interesting to add a discussion on whether there are direct use of the approach as it is, and how authors plan to take the temporal dimension in the future work.


Looking into the criteria specified in the track description I could not find plans for adoption and could not assess the impact. Regarding reproducibility, the authors plan to release the code after internal approval, but as of now the results are not reproducible.



Eiter, T., Füreder, H., Kasslatter, F. et al. Towards a Semantically Enriched Local Dynamic Map. Int. J. ITS Res. 17, 32–48 (2019). https://doi.org/10.1007/s13177-018-0154-x


**Anonymity:**

Yes, I would like my review to remain anonymous.

**Rating:**

-1: Weak Reject

**Strong Points:**


Clear use of SW technologies
Benefits over other classification methods clear


**Subreviewer:**

I submitted this review.

**Weak Points:**

Parts of the pipeline lack a deeper description (e.g. data ingestion, knowledge extraction, reasoning)
Plans for adoption and impact could not be assessed

---

> ### Author Rebuttal · Authors · 2021-01-28
>
> Thanks for the very useful comments and good points! Below, we try to answer each question:
>
> 1. Using ontologies on the ITS? Ontology in [17]? Local Dynamic Map-based ontology?
>
> Good suggestion about the ITS ontologies, we will explore it. Considering the licensing aspect, which is always a pain point in the industrial settings, we expect to be quite limited in the reusability.
>
> W.r.t. ontology in [17], though from the illustration we see several relevant concepts, unfortunately, we could not find the ontology itself so we could potentially reuse its concepts.
>
> The work in the suggested LDM paper is very interesting towards dealing with stream data. Authors cover important aspects on message exchanging between traffic participants. A rule-based mechanism is implemented provide question answering on top of stream data. Therefore, we see it as relevant for extending our approach for considering stream data and reasoning in a continuous learning scenario.
>
> 2. W.r.t Knowledge Extraction and reasoning?
>
> The Knowledge Extraction is focused on retrieving necessary information for the knowledge graph related- or downstream- tasks. Query Execution is the main component which in combination with a reasoner enables the execution of complex queries.
> The reasoning is used in cases like retrieving information based on the “occursBefore” property defined as inverse property of “occursAfter”. Also, in combination with Stardog rules to classify on the fly different RiskLevels or FirstDegreeImpact based on a given TimeToCollision (TTC) and some other parameters.  These aspects, remains to be analysed and exploited more in deep to have a better understanding of their effects. Thus, in this work we did not have mature insights and results that we considered interesting for reporting them.
>
> 3. Synthetic data with SUMO?
>
> Highly automated driving of at least ASIL Level 3 is of the research focuses of future cars [1]. The behaviour of HAD especially in critical driving situations is crucial in order to validate HAD systems and compare their performance with human drivers [2]. Validation based on recorded test drives [3] would require millions or billions of test kilometres to prove safety of HAD systems which makes them unfeasible for validation. Virtual testing is one main method to validate HAD systems [2]. SUMO is one of the main simulation tools for traffic simulation [4].
>
> [1]P. Koopman at al., “Challenges in Autonomous Vehicle Testing and Validation”
>
> [2]C. Wang at al, “Overcoming Challenges of Validation Automated Driving and Identification of Critical Scenarios”
>
> [3]N. Kalra at al, “Driving to safety: How many miles of driving would it take to demonstrate autonomous vehicle reliability”
>
> [4]P. Lopez, et al. "Microscopic traffic simulation using sumo"
>
> 4.	Mapping SUMO to CoSI?
>
> The SUMO output stored in 3 different files: 1) timesteps.xml (FCD output) - the scenery information and participant’s details in a scene; 2) conflicts.xml (SSM output) - conflicts between ego and a foe; and 3) lanechanges.xml  - lane change information. The content of these files is explained below:
>
> An excerpt from timestep.xml file:
>
> <timestep time="5.00">
>  <vehicle id="1" type="vehDist26" speed="5.9632" … /> ...
> </timestep>
>
> Each vehicle node represents a participant including its details and the RDF representation looks like:
>
> cosi:Scene_RL1_5 a cosi:Scene;
>  cosi:occursAfter cosi:Scene_RL1_4;
>  cosi:time "5"^^xsd:int;
>  cosi:hasParticipant [a cosi:Vehicle ;
>  cosi:id "1"^^xsd:integer; ...].
>
> One Conflict Scenario (CS) node has information for multiple Conflictual Points/Situations contained in fields like time step (field timeSpan) Situation type (field typeSpan has numbers representing various types of situations and is used to map to the ontological classes like FollowingFollower, FollowingLeader), as described in the following conflicts.xml excerpt:
>
> <conflict begin="21" end="22" ego="2" foe="8">
>  <timeSpan values="21 22 ..."/>
>  <typeSpan values="10 10 .."/>
>  <minTTC time="23" position="6.93,50.96" type="11" value="1.81"/> …
> </conflict>
>
> We extracted the metadata like: beginTime, endTime, ego, foe, minTTC values to create CS individuals:
>
> cosi:CS_RL1_21_28_102_2_8 a cosi:ConflictScenario;
>  cosi:beginTime "21"^^xsd:integer;
>  cosi:endTime "28"^^xsd:integer;
>  cosi:egoID "2";
>  cosi:minTTC "22"; ... .
>
> Next, conflictual points are extracted by splitting the values in each field using “space” as delimiter and iterating over them to create RDF individuals as below. New relationships like the scene in which a conflict occur and part of which CS are established:
>
> cosi:C_RL1_102_21_2_8 a cosi:CrossingLeader;
>  cosi:time  "21"^^xsd:integer;
>  cosi:occursIn cosiData:Scene_RL1_102_21;
>  …
>  dul:isPartOf cosiData:CS_RL1_21_28_102_2_8 .
>
> cosi:C_RL1_102_22_2_8 a cosi:CrossingLeader;
>  cosi:time  "22"^^xsd:integer;
>  cosi:occursIn  cosiData:Scene_RL1_102_22;
>  ….
>  dul:isPartOf cosiData:CS_RL1_21_28_102_2_8 .
>
> The same principle is applied for lane changes.

---

### Official Review · AnonReviewer5 · 2021-01-17
**Review of article "Towards a Knowledge Graph-based Approach for Situation Comprehension in Driving Scenarios"**

**Rating:** 1
**Confidence:** 3

**Review:**

The authors describe a knowledge-graph-based approach for representing data related to traffic situation; the proposed approach is based on a custom ontology. The authors evaluate their approach in a classification task using knowledge graph embedding, and compare the results against vector-based classification techniques.

The article is well written, and interesting.

Pros:
- The authors clearly describe the domain, and define the problem.
- The technical description of the approach is clear.

Cons:
- Not sure if the evaluation is comprehensive: is the classification task representative of real world problems?
- The proposed approach is evaluated on simulated data: how different is the simulated data compared to real-world data? Will the proposed approach work on real-world data?
- Not sure that the evaluation is technically accurate - see detailed comments below.

Note:
Rating updated to "Weak Accept" after reading the rebuttal.

**Anonymity:**

Yes, I would like my review to remain anonymous.

**Strong Points:**

- The authors clearly describe the domain, and define the problem.
- The technical description of the approach is clear.

**Subreviewer:**

I submitted this review.

**Weak Points:**

- Section 4.2: "Once the transformation process is realized": which transformation process?
- Section 4.2, Ontology: Scene, Situation, Scenario: what is the contribution/novelty of this article with respect to [33]?
- Section 4.2, Transformation: "Raw data generated from sensors should be augmented with additional semantic information.": why? Please elaborate.
- Section 4.2, Transformation: "In these cases, a number of additional queries are executed on-the-fly to enrich sensor data with new relationships.": what queries? On which data?. Similarly in the next paragraph "Knowledge Extraction"
- Section 5: "Models that consider temporal in- formation are very likely to lead to better results": why? Also, adding the temporal dimension would most likely increase the computational complexity: how will you handle that?
- Table 2: TTC?
- Section 5.4: It is not clear which features you have extracted from the KG, and how you chose them.
- Section 6.1: "For our experiments we used a sub-set of the CKG with 1.4 million triples divided in 134K Conflict Scenarios for training, 46K for validation and 46K for testing.": 134K + 46K + 46K != 1.4M ... can you please explain?
- Section 6.1: "Various experiments are performed for single features, all features and the 5 most important features.": which are the 5 most important features? How did you select them? Why are the most important?
- Section 7: not very informative. You may want to remove it if you need space to provide more details on the evaluation.
- Some minor language problems:
  - Abstract: "... achieve 95% accuracy _whereas_..."
  - Section 1: "Vehicles with driver assistance systems (DAS) aim to take some work-load _off_ ..."
  - Section 4, first line: I think you should remove "communicating"
  - Section 4.2: "Comprehending situation requires integrating and structuring the abundance of information from different sources before being ready to be used": I suggest that you remove "before being ready to be used"
  - Section 5: "... where vehicles base their behavior _on_ actual and ..."
  - Section 5.1: "It is based on the assumption that the speed of the ego vehicle is adopted according ...": adopted -> adapted?

---

> ### Author Rebuttal · Authors · 2021-01-28
>
> We appreciate your intensive review and valuable feedback.
>
> 1. classification task representative of real world problems?
>
> 2. simulated vs real-world data?
>
> Highly automated driving is a main research focuses of future cars[1]. The behavior of HAD especially in critical driving situations is crucial in order to validate HAD systems[2]. Validation based on recorded test drives[3] would require billions of test kilometers which makes it unfeasible for validation. Virtual testing is an accepted method to validate HAD systems[2] and SUMO is one of the main simulation tools[4].
>
> [1]Koopman at al-Challenges in Autonomous Vehicle Testing and Validation
>
> [2]Wang at al-Overcoming Challenges of Validation Automated Driving and Identification of Critical Scenarios
>
> [3]Kalra at al-Driving to safety: How many miles of driving would it take to demonstrate autonomous vehicle reliability
>
> [4]Lopez, et al-Microscopic traffic simulation using sumo
>
> 3.working on real-world data?
>
> HAD vehicles will provide signals similar with those in SUMO. We assume that our approach will work on real-world data, although with slightly lower performance due to perception errors. Virtual assessment could validate HAD system as described in [1].
>
> [1]Wachenfeld et al-Virtual Assessment of Automation in Field Operation A New Runtime Validation Method
>
> Weak points:
>
> 1.which transformation process?
>
> We agree this part should have some explicit statements to make it clear for the reader. The transformation process is related to converting input data of any format to RDF triples, i.e. XML data to RDF triples.
>
> 2.contribution/novelty w.r.t. [33]?
>
> [33] is primarily focused on providing a generic definition for concepts of Scene, Scenario and Situation, respectively from the terminology point of view and describing their respective elements that compose each of them. We used these descriptions to create formal definitions using ontological axioms. We defined additional concepts such as Driver (personal characteristics), Car Occupants, Observation, Geospatial Information, etc., as well as their inter-relations.
>
> 3.w.r.t. augmentation?
>
> We augmented raw data with semantic information like type definition (Situation types generated as a numbers in SUMO are mapped to respective classes), new relationships (see the next point), data-types: geospatial points. For instance, utilizing geo:wktLiteral, we experimented on retrieving all conflicts occurring in a given radius specified in miles/kilometres based on UnitsOfMeasure Ontology and SPARQL geospatial functionalities.
>
> 4.enrichment via queries?
>
> SUMO data is split based on the predefined Driving Scenarios. Each Driving Scenario has several so-called Examples with multiple time frames. Information for each Example is further split in 3 different files: 1) scenery information; 2) conflicts; and 3) lane changes. There is no concrete relationship between these files, apart from the name of Driving Scenario, Example Id and the timestep. These 3 components are used in queries to define unique IRI’s for each time-step. The new formed IRIs are then used to establish relationships between scenes, conflicts and lane changes. Since there is no relationship of scenes apart from time consequence, we created new relation (“occursAfter”) between consecutive scenes using time info. The “occursBefore” is defined as inverse, so the actual triples are not materialized but can be inferred. We defined different classes of Situations based on: shorturl.at/glAOS
>
> 5.modelling temporal information? and the impact on performance?
>
> Using temporal information could improve the results because some situations such as merging or overtaking are manifested by their gradual change of the lane ID over time. The current approach takes about 0,239 msec test time per sample. We are working on including temporal information and expect slightly higher computation time.
>
> 4.TTC?
>
> TTC: Time to collision, the time until a collision between two entities would occur if both continue with the present velocities.
>
> 5.KG extracted features?
>
> We used the same entities and literal parameters as in the KG experiments, except the relation information. For each vehicle, we have about 10 parameters (e.g. position, speed, acceleration, steering angle). The number of other parameters (e.g. minTTC, velocity difference etc.) depends on the number of vehicles in the vicinity (50m radius) around ego.
>
> 6.Scenarios and Triples discrepancy?
>
> Actually, here is a mistake; the 1.4M should be 6.4M, representing around 226K Conflict Situations (CS), where 134K are used for training, 46K for validation and 46K for testing. Each CS is described with around 28 triples in average, so 226K x 28 ~= 6.4M.
>
> 7.five most important features and their selection?
>
> These are shown in Table 3 and are selected based on heuristics about the driving situations, e.g., features such as the vehicle ID, maximum speed, acceleration are unlikely to contribute and were not included.

---

> > ### Comment · AnonReviewer5 · 2021-02-09
> > **Answer to Rebuttal**
> >
> > Thank you for your explanations and clarification. I strongly encourage you to update your article and include the explanations/clarifications that you provided in the rebuttal, particularly 1, 2, 3, 4, 5, and 6.

---

### Decision · Program_Chairs · 2021-02-23

**Decision:**

Accept with shepherding

**Comment:**

All the reviewers agree that the approach is a very good solution for situation comprehension in driving, based on semantic web technologies. While the approach has been very well implemented, and a comprehensive evaluation is provided, as it has been acknowledged by the recommendations provided by the reviewers, the authors do not provide a clear evidence how this approach is being used in a real-world application and what is the target user base, which is extremely important for the in-use track.

Therefore, we propose this paper as an accept with shepherding, so that we can make sure that in case that this work is being actually used, we can include the necessary paragraphs in the paper so as to demonstrate this fit. If in the end this paper is not suitable for the in-use track, it may be good to consider it for the industry track, which will open soon.